# The maximal calber method for the evolution models, and infintely complex adaptive systems

David B. Saakian [1*]

[1] *A.I. Alikhanyan National Science Laboratory (Yerevan Physics Institute) Foundation,*
*2 Alikhanian Brothers St., Yerevan 375036, Armenia*

(Dated: February 6, 2026)

We consider the maximal caliber method from chemistry and stochastic thermodynamics, applying it to the evolution. We investigated the Wright Fisher model, described via Fokker-Planck equation, introducing the Maximal Caliber. The active inference model is also described by Fokker Planck equation, so our results can be extended for this case as well. While in Active Inference literature has been discussed the Maximal Caliber method, they did not derive the expression for the Caliber function. We identify the class of infinitely complex adaptive systems (ICAS), including the evolving complexity during the evolution, the consciousness and AGI. We assume a similarity for the main concepts and math tools for the solution of all the members of this universality class.

## I. INTRODUCTION

A possible connection of stochastic thermodynamics with the evolution, first considered in [1] in relation of fluctuation theorems, is a rather hot topics related mainly with fluctuation relations [2],[3],[4],[5]. We need in the advanced statistical physics as the standard statistical physics can noot be applied directly in evolution, due to non monoton behaviour of entropy [6].Another activity, related to stochastic thermodynamics is the Maximal Calibier method [7], suggested 4 decades ago by Jayens [8], looking the sum via the trajectories [9] under some constraints. Actually the Max Cal can be considered as a simple generalization of the maximal entropy principle. Another related concept is the theory of functional systems by Anokhin [10]. There is some goal, and the physiology is optimesed to achieve it. In case Maximal Calibier [12–15], we assume an existence of some fluxes, then solve the model assuming a maximal entropy under that observation. The concept is somehow related to the optimal control as well. Close ideas have been suggested in [17], where the authors try to understand the biological data as a randomness under some constraints.

In the current article we are planning to formulate the Max Cal for the evolution models, as well as discuss the idelogical aspects of the model. Compared to other optimization methods, Max Cal is focused on the currents, and nonzero currents, especially related to some topological forces [18] are one of the characteristics of the life.

The Max Cal has been formulated for the Markovian models, which are the simplest version of the linear Master equation. In the current work we formulate first the model for the quasispecies evolution model which is the simplest version of the nonlinear master equation, then for the Fokker Planck equation, the second (afther the Markovian model) favorite model of stochastic thermodynamics. Later we introduce the Max Cal for the 2 level selection evolution model. The case of nontrivial hierarchy is especially interesting for us, as the goal can be formulated for the top level of hierchy only.

______

\* saakian@yerphi.am

## II. MAXIMAL CALIBER FOR THE MARKOV MODEL

Consider first the known results for the Maximal Calibier method for Markovian models.

### A. Discrete time Markov model

The maximal caliber method has been discussed in details in [12],[13],[14] on the basis of [9]. We will briefly repeat the results of [12],[13],[14].

Consider the random walks with the transition matrix $k_{ij}$. The probability of the sequence is

$$P(C) = p_{i_0} k_{i_0,i_1} k_{i_1 i_2} ... k_{i_{s-1},i_s} \quad (1)$$

After the long period of relaxation, $s \gg 1$ we get the following formula

$$P(C) \sim \exp(-sH),$$
$$H_0 = \sum_{i,j} p_i k_{ij} \ln(k_i j) \quad (2)$$

where $p_i$ is the steady state distribution.

Eq.(2) can be derived as an entropy of the distribution via the trajectories.

We can put some constraints, $F_l(p,k) = 0$ defined on the trajectories, then in that case we should look for the extremum

$$H = H_0(p,k) + \sum_l X_l F_l \quad (3)$$

actually we can put the constraints for every moment of time.

Let us put the constraints.

$$H = -\sum_{i,j} p_i k_{ij} \ln(k_{ij}) + \gamma \sum_{i,j} (p_i k_{ij} r_{ij} - r) +$$
$$\sum_i n_i p_i (\sum_j k_{ij} - 1) + \sum_j m_j (\sum_i p_i k_{ij} - p_j)$$
$$+ \lambda (\sum_{i,j} p_i k_{ij} - 1) \quad (4)$$

where we used the probability balance condition

$$\sum_{ij} p_i k_{ij} = 1 \tag{5}$$

$$k_{ab} = \exp(n_a + m_b - 1 + \lambda - \gamma r_{ab}) \tag{6}$$

$$n_a + m_a = 1 \tag{7}$$

define

$$Q_{ab} = \exp(-\gamma r_{ab}), \\ \eta = e^{-\lambda} \tag{8}$$

and define the eigenvalue problem

$$\sum Q_{ab}\xi_b = \eta \xi_a, \\ \xi_a = \exp(-n_a) \tag{9}$$

and

$$k_{ab} = \exp(n_a - n_b + \lambda - \gamma r_{ab}) \tag{10}$$

We have the eigenvector equation

$$\sum_b f_b Q_{ab} = \eta f_a \tag{11}$$

Then the steady state distribution $p_a$ provides

$$\sum_a \frac{p_a}{f_a} Q_{ab} = \eta \frac{p_b}{f_b} \equiv \xi_b \eta \tag{12}$$

Thus

$$p_a = \xi_a f_a \tag{13}$$

Thus we have

### B. Continuous time Markov model

Consider the limiting case

$$k_{ij} = \epsilon m_{ij}, i \neq j \\ k_{ii} = 1 + \epsilon m_{ii} \tag{14}$$

We have for the path entropy

$$-\epsilon \sum_i p_i m_{ii}) - \epsilon \sum_{i,j} p_i k_{ij} \ln(m_{ij}) \tag{15}$$

Then for the optimization function

$$H = -\epsilon \sum_i p_i m_{ii}) - \epsilon \sum_{i,j} p_i m_{ij} \ln(m_{ij}) \\ + \gamma \sum_{i \neq j}(p_i k_{ij} r_{ij} - r) + \\ \epsilon \sum_i n_i p_i (\sum_j m_{ij} - 1) + \epsilon \sum_j m_j (\sum_i p_i m_{ij} - p_j) \\ + \epsilon \lambda (\sum_{i,j} p_i m_{ij} - 1) \tag{16}$$

We get the optimization equations

$$-(\ln m_{ij} + 1) + \gamma r_{ij} + n_j + m_i = 0, i \neq j \\ -1 + n_i + m_i = 1, \\ -m_{ii} - \sum_{ij} m_{ij} \ln m_{ij} + \gamma \sum_{i \neq j} m_{ij} r_{ij} + \\ \sum_{i \neq j} m_{ij} n_j \frac{x_j}{x_i} + \lambda = 0 \tag{17}$$

We get from the system of equations

$$m_{ij} = \exp(\gamma r_{ij} + n_j - n_i), i \neq j \\ m_{ii} = \lambda \tag{18}$$

Then the probability balance condition $\sum_j m_{ij} = 0$ gives

$$A_{ij} = e^{\gamma r_{ij}} \delta_{i \neq j} + \lambda \delta_{i,j}, \\ A.x = 0, \\ x_i = \exp(-n_i) \tag{19}$$

Then

$$\xi.A = 0\xi \tag{20}$$

and

$$p_a = x_a \xi_a \tag{21}$$

## III. WRIGHT FISHER MODEL

Consider the Wright Fisher model. We heve d genome types in the population with the integer numbers $n_1...n_d$, where $n_1 + ..n_d = N$. We define the continuous variables $x_i = n_i/N$, and functions

$$w_i = X_i + f_i(X) \tag{22}$$

The iteration rule is

$$p(j_1...j_d) = \frac{N! w_1^{j_1}..w_d^{j_d}}{j_1!...j_d} \tag{23}$$

### A. Known results

For the large N and proper choice of $f_i(x)$ functions (small selection mutation coefficients), we get the infinite population dynamics

$$\frac{dX_i}{dt} = f_i(X) \qquad (24)$$

The steady state point is $X_i^0$.

For the finite population case we have for the probability distribution $p(x,t)$

$$\frac{dp}{dt} = -\sum_i \frac{d}{dX_i} f_i(X)p(X,t) + \alpha \sum_i \frac{d}{dX_i} D_{ij} \frac{d}{dX_j} p,$$
$$D_{ii} = X_i(1-X_i),$$
$$D_{ij} = -X_i X_j \qquad (25)$$

Let us define $x_i = X_i - X_i^0$. We define

$$M_{ij} = \frac{df_i}{dx_j}\Big|_{x_i=0} \qquad (26)$$

At the steady state

$$P(x) = \exp[-\frac{T_{ij}x_i x_j}{2\alpha}] \qquad (27)$$

Then we have an equation [19]

$$M.T + T.M + T.g.T/2 = 0 \qquad (28)$$

Then we have for the Entropy Production rate [19],[4]

$$\sigma = -\int dx Tr J \frac{1}{\alpha D} J = Tr \frac{1}{T}(A - \frac{gT}{2})\frac{1}{g}(A - \frac{gT}{2}) \qquad (29)$$

### B. Max Cal for the WF model

There are many possibilities putting the constraint. Let us put a constraint on a EPR. Thus we can consider the optimization

$$Tr \frac{1}{T}(A - \frac{gT}{2})\frac{1}{g}(A - \frac{gT}{2})^* \qquad (30)$$

$$H = -\int dy p(y)\rho(x,y) \ln \rho(y,x) + h = d - d\ln \epsilon +$$
$$\frac{1}{2}\ln(det(g)) + \lambda(Tr \frac{1}{T}(A - \frac{gT}{2})\frac{1}{g}(A - \frac{gT}{2})^* - c)$$

Thus we can take as a caliber function

$$H = \frac{1}{2}\ln(det(g)) + \lambda(Tr \frac{1}{T}(A - \frac{gT}{2})\frac{1}{g}(A - \frac{gT}{2})^* - c)$$

## IV. THE MATH TOOLS FOR THE SOLUTION OF INFINTELY COMPLEX ADAPTIVE SYSTEMS

We intuitively choose 3 Complex adaptive Systems with an infinite complexity (ICAS):
The birth of life, growing complexity during the evolution,
The consciousness,
3. AGI.
We assume a hypothesis that all 3 belong to the same universality class, so should share the same key properties, the methods of investigations. Among the methods are: Stochastic thermodynamics (SC), including the Maximal Caliber (Max Cal) method. Among the key features are the vortices (monopoles) and at the top level Classical versions of quantum anomalies (Nigel Goldenfeld, A. Abanov, P. Wigman). Without the solving the latter, we can not solve any of mentioned problems. Every biologist should agree that both the life, consciousness are the nonzero currents. Then SC,Max Cal, anomalies just math tools for the currents. In the current work we construct the Max Cal for 1 and 2. By the way, I met Max Cal first in AGI literature , in the article by Ben Goertzel, he was claiming that the method can bring to the solution of AGI.

Next, to understand the ICAS systems, we need to solve the classical version of quantum anomalies. There are few works about this subject [22],[23]. Another important direction is the solution of classical entanglement problem. In evolution we are sure that the complexity growth is related with the frustration of selection between different hierachy levels. We have a cyclic logics in the models of consciousnees by [21] and [20], somehow related to the vortes structure in stochastic thermodynamics. I amoptimistic that the advance in mentioned 3 direction together can bring to the solution of all three problems, including AGI.

## V. CONCLUSIONS

Max Cal method successfully has been applied in chemistry and gene expression models, using the linear Master equation, the Markovian models. In the current work we extended the method for the nonlinear master equation: the (nonlinear) evolution equations. The method is becoming much more involved. Next, we extended the method for the Fokker Planck equation, the case of Wright-Fisher evolution model, when the observed current is related to the entropy production rate. We looked at the simplest case of point attractor. We can extend our results, observing via Caliber method the general cases of the fluxes, as well as looking the limiting cycle case. The latter construction can be interesting for the Fokker-Planck equation, related to the Free Energy Principle by K. Friston [20]. We have 4 dimensional FP equation there.

We can apply the [16] method to calculate the gene expression network, as well as apply it to the Swarm intelligence, for the initial and final configurations. Next, the Anokhin's principle of functional systems resembles the Maximal Caliber principle as well, and it will be very interesting to analyze the functional systems using the latter. Anokhin defines

t system as something, where all the iteractions betwee the parts are defined for the getting the useful result: so we should look the maximum entropy via trajectories with a given result. We need to analyze the models by Vityaev from MC perspectives as well. The last, we can look the emotions (the person should change his emotional state) and active inference (the flux means change the states of 4 components) from the MC calibier under the given flux.

He we suceed well with the Max Cal, then discussed 2 another methods as well: the classical version of quantum anpmalies, and the classical version of entanglement. We assume a good progress in all 3 directions can bring to the solution of AGI as well.

I thank K. Friston for discussions. The work has been supported by the HESC of Armenia, grants No. 24IRF/2-1C001, 10-27/24FP-1D009.

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
