# OpenReview forum: "The maximal calber method for the evolution models, and infintely complex adaptive system"
_mathai.club/MathAI/2026/Conference — MathAI 2026 Conference Submission_

### Official Review · Reviewer_q9Bo · 2026-03-11
**Interesting idea, but the paper is too speculative and insufficiently substantiated**

**Rating:** 4
**Confidence:** 1

**Review:**

\textbf{Summary.} The paper explores an interesting direction: applying the Maximum Caliber framework to evolutionary models, in particular to the Wright--Fisher model via a Fokker--Planck formulation. It also discusses a much broader class of ``infinitely complex adaptive systems'' including consciousness and AGI.

\textbf{Strengths.} The general idea of connecting MaxCal, stochastic thermodynamics, and evolutionary dynamics is potentially interesting. A MaxCal formulation for Wright--Fisher-type models could be valuable if developed rigorously.

\textbf{Weaknesses.} In its current form, the paper is too speculative and not sufficiently rigorous. The mathematical derivations are presented too briefly, with several notation and typographical issues that make verification difficult. Important quantities and transitions between formulas are not defined clearly enough. In addition, the paper makes a very large conceptual jump from the Wright--Fisher / MaxCal setting to broad claims about infinitely complex adaptive systems, consciousness, and AGI, without adequate technical support. Finally, there is no numerical or empirical validation.

\textbf{Recommendation.} Overall, the paper contains an interesting idea, but in its present form it reads more like a speculative research note than a complete conference paper. I therefore place it below the acceptance threshold.

---

### Official Review · Reviewer_mLxC · 2026-03-12
**The paper proposes a new universality class called "Infinitely Complex Adaptive Systems" (ICAS) to group biological evolution, consciousness, and Artificial General Intelligence**

**Rating:** 6
**Confidence:** 4

**Review:**

Summary: The paper attempts to apply the Maximal Caliber method (a variational principle from stochastic thermodynamics) to biological evolution models, specifically the Wright Fisher model. The author derives expressions for the Caliber function under certain constraints and proposes a new universality class called "Infinitely Complex Adaptive Systems" (ICAS) to group biological evolution, consciousness, and Artificial General Intelligence (AGI). The author argues that methods from stochastic thermodynamics and classical analogues of quantum anomalies are necessary to solve problems in all three domains.

Strengths: Theoretical Foundation: The derivation of the Caliber function for the Markov and Wright Fisher models in Sections II and III follows sound principles of statistical physics.
Ambitious Scope: The paper attempts to bridge disparate fields (biology, thermodynamics, AI) with a unifying mathematical principle.

Weaknesses: Lack of Rigor in Key Claims: The core claims regarding ICAS, consciousness, and AGI are highly speculative and lack mathematical substantiation. The jump from population genetics calculations to asserting a solution path for AGI is unsupported.
Limited Relevance to AI: The paper does not provide any actionable methodology for AI research. It frames AGI as a theoretical problem to be solved by physics tools but offers no concrete results or implementations relevant to AI.
Disconnect: The link between the specific calculations performed (Wright Fisher model) and the broad philosophical conclusions is weak.
Poor Presentation: The manuscript is riddled with typos and grammatical errors (e.g., "calibier" in the title and text, "heve", "lnot"), significantly hindering readability.

---

### Official Review · Reviewer_k37j · 2026-03-12
**Interesting research direction but the conclusions are not sufficiently supported by mathematical proofs.**

**Rating:** 6
**Confidence:** 1

**Review:**

The author considers the Maximum Caliber method (MaxCal) from chemistry and stochastic thermodynamics, applying it to the evolution of complex systems. Previously, MaxCal has been successfully applied in chemistry and gene expression models, using the linear Master equation, the Markovian models. The author claims to extend the Maximum Caliber method to the nonlinear master equation: the nonlinear evolution equations. Next, he extends the method for the Fokker-Planck equation, as a special case of Wright-Fisher evolution model, looking at the simplest case of point attractor. He also claims that his results can be extended to treat the active inference model, described by Fokker Planck equation, but this claim is made only in the abstract and in conclusions, without any further details. The author claims that he has identified the class of infinitely complex adaptive systems (ICAS), including the evolving complexity during the evolution, the consciousness and AGI. But there is no strict definition of this class in the paper, just a list of these three examples.

In general, the paper uses the rigorous mathematical language, but makes transitions between different propositions, which are hard to verify. Either the author should extend the paper to fill the gaps or refer to the relevant literature. Currently the mathematical foundation of the new results is concentrated in few lines of Subsection III.B, which is clearly insufficient to support all the claims made by the author.

Nevertheless, I hope that the author will be able to fill most of the gaps, if the paper is accepted.

Minor remarks

1. The term “Maximum Caliber” appears in multiple spellings and misspellings, even in the title.
2. Subsection II.B ends with the words “”Thus we have”. I guess the conclusion is missing here.
3. Formula (23) has a misprint: it should be j_1!...j_d! In the denominator, I guess.
4. In the beginning of Section IV, the author is going to list three examples of ICAS. But I see four of them here: (1) The birth of life, (2)  growing complexity during the evolution, (3) The consciousness, (4) AGI.
5. The term "current" has not been introduced in this paper. A reader unfamiliar with MaxCal theory has no chance to understand it when it appears on page 3.
6. The paper is not anonymous, as required.

---

### Decision · Program_Chairs · 2026-03-20

**Decision:**

Accept (Oral)

**Comment:**

Dear Author(s),

On behalf of the Program Committee of the International Conference on Mathematics of Artificial Intelligence (MathAI 2026), we are pleased to inform you that your paper has been accepted for an oral presentation at MathAI 2026.

Your paper was evaluated through a rigorous two-stage review process involving both automated screening and expert review by members of the Program Committee. The reviewers recognized the quality and contribution of your work.

Presentation details:

- Format: Oral presentation (15–20 minutes + 5 minutes Q&A)
- Mode: You may present either in person (offline) at the conference venue in Sirius, Russia, or remotely via Zoom. Please indicate your preferred mode when confirming your participation.
- Conference dates: Marh 30 - April 3, 2026
- Website: https://mathai.club

Next steps:

1. Please confirm your participation and presentation mode by replying to this email mathai.club@yandex.ru no later than March 15, 2026 18:00 Moscow time.
2. If you plan to attend in person, the organizing committee will provide accommodation details separately.
3. Please prepare your final camera-ready manuscript according to the formatting guidelines available at https://mathai.club and upload it to OpenReview by March 15, 2026 18:00 Moscow time.

Should you have any questions regarding the program, logistics, or your presentation slot, please do not hesitate to contact us.

We look forward to your contribution to MathAI 2026.

With kind regards,

MathAI 2026 Program Committee
International Conference on Mathematics of Artificial Intelligence
https://mathai.club
OpenReview: https://openreview.net/group?id=mathai.club/MathAI/2026/Conference
MathAI Telegram: https://t.me/MathAI_club
IAIC International AI Committee: https://t.me/iaic_world
Email: mathai.club@yandex.ru